# FORECASTING DEEP LEARNING DYNAMICS WITH APPLICATIONS TO HYPERPARAMETER TUNING

## ABSTRACT

Well-performing deep learning models have enormous impact, but getting them to perform well is complicated, as the model architecture must be chosen and a number of hyperparameters tuned. This requires experimentation, which is time-consuming and costly. We propose to address the problem of hyperparameter tuning by learning to forecast the training behaviour of deep learning architectures. Concretely, we introduce a forecasting model that, given a hyperparameter schedule (e.g., learning rate, weight decay) and a history of training observations (such as loss and accuracy), predicts how the training will continue. Naturally, forecasting is much faster and less expensive than running actual deep learning experiments.

The main question we study is whether the forecasting model is good enough to be of use - can it indeed replace real experiments? We answer this affirmatively in two ways. For one, we show that the forecasted curves are close to real ones. On the practical side, we apply our forecaster to learn hyperparameter tuning policies. We experiment on a version of ResNet on CIFAR10 and on Transformer in a language modeling task. The policies learned using our forecaster match or exceed the ones learned in real experiments and in one case even the default schedules discovered by researchers. We study the learning rate schedules created using the forecaster are find that they are not only effective, but also lead to interesting insights.

## 1 INTRODUCTION

Machine learning researchers working with deep neural networks spend time looking at training curves and asking themselves the question: How would the results change if I modified some hyperparameters? They run many experiments and, with time, develop a better understanding of how modifications in various hyperparameters affect the learning dynamics of our models.

In this work, we attack this problem of understanding deep learning dynamics with deep learning itself. To this end, we collect a data-set of training curves of deep learning models with a diverse set of hyperparameter schedules. Next, we train an autoregressive deep model to predict the training curves conditioned on the schedule.

More concretely, our model observes the setting of a small number of hyperparameters, such as learning rate, weight decay, or dropout rate. Then, it predicts a small number of values that a researcher would usually look at, such as training and validation loss and accuracy. We predict how these values will change a few hundred training steps later. Then, the model gets new settings for the next few hundred training steps, and predicts the next values. The model is autoregressive, so it uses the history of its own predictions from previous time-steps to predict the next ones.

We study how to do time-series prediction with autoregressive models even in the presence of stochastic behaviour. We study different losses and introduce a loss weight scheme that allows to use Transformer

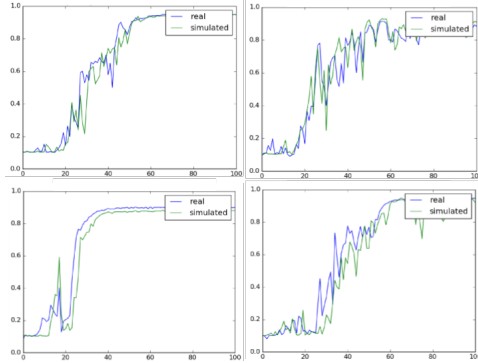

Figure 1: Real and predicted evaluation accuracy curves, see text on the left for details.

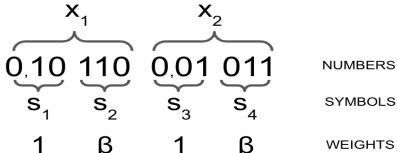

Figure 2: Discretization procedure where $x_t$ is the discretized sequence. It is represented in fixed-precision encoding with 2 base-8 digits per number, so each symbol $s_i$ corresponds to 3 bits of precision. Weights of symbols are decayed with parameter $\beta$ according to their significance, so the more significant digit has weight 1 and the other has weight $\beta$.

models for stochastic time-series predictions. We show that the new method we introduce outperforms baselines and is able to faithfully approximate the learning dynamics of deep learning models, as shown in the examples in Figure 1.

Having a model of learning dynamics is interesting in its own right, as it allows to study the training process much faster, without running full experiments. But it also opens up the possibility to learn hyperparameter schedules by using reinforcement learning in the model. We experiment with this and validate our forecaster: the policies learned using reinforcement learning and the forecaster work well when training real models. Running the forecaster is of course much faster than running real experiments, so learning a policy in the forecaster is thousands of times faster.

## 2 TIME-SERIES MODELING WITH TRANSFORMER

Transformer (Vaswani et al., 2017) is an autoregressive neural architecture for discrete sequence prediction and transduction. Its variants achieve state-of-the-art results in machine translation and language modeling (Shoeybi et al. (2019), Radford et al. (2019)). The architecture is notably composed almost entirely of self-attention layers, with no recurrent connections. This allows the model to consider the entire sequence when predicting the next token, while being very efficient to train using GPUs, due to the massive parallelism allowed by attention layers.

In this work, we show that Transformer can also be used to predict time series – concretely, training curves of deep learning models. One obvious way to apply Transformer to time-series prediction would be to try to directly predict the next value in the sequence with L2 loss. We show that this is not a good strategy when sequences can be stochastic and introduce a discretization technique that reduces the problem to the standard discrete sequence prediction setting and that yields good results.

### 2.1 DISCRETIZATION

In order to model time series using Transformer, we represent each number in the sequence in fixed-precision, base-$k$ encoding. We concatenate the consecutive base-$k$ digits corresponding to each number, starting from the most significant one. To predict multiple time series at a time, we arrange the representations of a single element of each time series in a fixed order, and then concatenate the representations of each time step, as shown at Figure 2. This way we can model the joint probability distribution of multiple time series using a single model. We train the Transformer decoder to predict consecutive symbols in this discrete sequence autoregressively. We use cross-entropy as the loss function, weighted with exponential decay according to the significance of each digit:

$$L(t, p) = \sum_{i=0}^{n-1} \beta^i H(t_i, p_i)$$

where $H$ denotes the cross-entropy, $t_i$ are the true digits, $p_i$ is the model prediction and $i$ indexes digits in the fixed precision, base-$k$ encoding of a number, starting from the most significant one.

### 2.2 RESULTS ON SYNTHETIC DATA

We present an experiment on synthetic data, demonstrating the ability of Transformer to model a distribution over time series. The experiment is simple by design, devised so that it is easy to measure

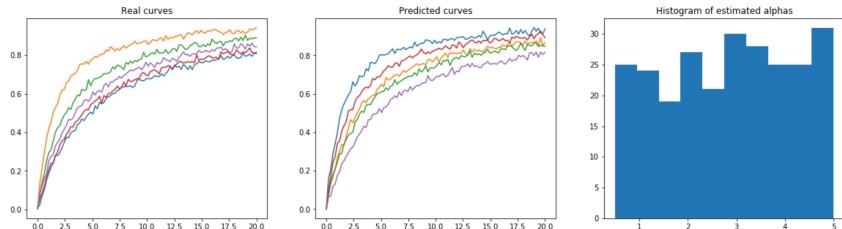

Figure 3: Synthetic curves generated by discrete Transformer decoder with weighted cross-entropy loss together with the histogram of estimated $\alpha$ values. In the generation process we used a uniform distribution and the histogram approximates it well.

the diversity of data sampled by the model, while retaining high-level similarity with the real-world data. Diversity in our case is crucial because, as described in the next section, the learned model is used to provide experience for training an reinforcement learning agent. We want this experience to be as diverse as in the real environment, so the agent can learn to be robust to various future scenarios.

We train Transformer on a dataset of synthetically generated curves calculated using the formula:

$$x_i = 1 - \frac{1}{1 + \frac{i}{\alpha}} + \mathcal{N}(0, \sigma^2)$$

where $i$ is an integer from the interval $[1, N]$ and $\sigma$ is scale of the noise, set to $0.01$. The dataset is designed to mimic training accuracy curves, starting from 0 and converging to 1 in the limit. Rate of convergence is controlled by $\alpha$, sampled uniformly from the interval $[0.5, 5]$ for each curve.

To measure the diversity of the generated curves, we estimate the parameter $\alpha$ from a curve predicted by the model, by averaging over pointwise estimates along the curve:

$$\hat{\alpha} = \frac{1}{N} \sum_{i=0}^{N-1} t(\frac{1}{x_i} - 1)$$

We then visualize the distribution of $\hat{\alpha}$ in a histogram. As shown in Figure 3, Transformer is able to generate diverse and convincingly-looking curves.

As a baseline, we consider modeling the curves without discretization, using the Transformer decoder without embedding and predicting a sequence of real numbers, training the model with L2 loss. During inference, we add Gaussian noise with variance $\sigma^2$ to the prediction at each step, to introduce stochasticity. As shown at Figure 4, the curves generated this way tend to collapse to a single shape.

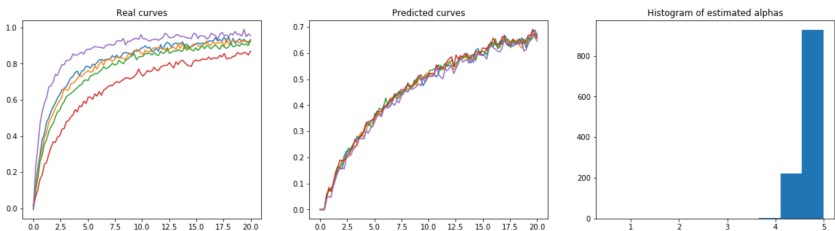

Figure 4: Synthetic curves generated by continuous Transformer decoder with L2 loss together with the histogram of $\alpha$ values that shows collapse.

## 3    HYPERPARAMETER TUNING USING REINFORCEMENT LEARNING

As is standard in reinforcement learning, we frame the problem of controlling model hyperparameters during training as a partially-observable Markov decision process. Each transition in the MDP corresponds to $n$ steps of training followed by an evaluation on a held-out set of $5\%$ of the training

data. Observations are the current values of training and evaluation accuracy and loss. Note that the agent does not have access to full state of the environment, because that would encompass the current values of all parameters of the model, which is intractable as an input for the agent. Rewards are differences between the values of an optimized metric between two timesteps, so that the cumulative reward is equal to the final value of that metric. In all experiments we optimize for validation accuracy.

Actions are relative changes in each controlled hyperparameter. Instead of predicting continuous actions using a parametric policy, we use a simple discretization scheme. Discretization of the action space allows the policy distribution to be multi-modal and has been empirically shown to improve training of reinforcement learning agents in terms of average performance, stability and robustness to hyperparameters (Tang & Agrawal, 2019). Concretely, we predict the change independently for each hyperparameter, out of a discrete set of values $\{-50\%, -20\%, -5\%, 0, +5\%, +25\%, +100\%\}$. The action space conveys the intuition that a hyperparameter schedule should be approximately continuous in the log space, while allowing for both significant changes in the hyperparameters between timesteps, and for more fine-grained control. Opposite values of relative change cancel each other out, so that a random walk in the action space has a median relative change of approximately 1 for every hyperparameter.

### 3.1 Model-free approach

As a basic model-free approach to solving this problem, we use the Proximal Policy Optimization algorithm (Schulman et al., 2017). PPO is a widely-used policy gradient method, notable for being stable and easy to implement, while significantly improving upon the sample-efficiency of vanilla policy gradient. It allows to perform multiple gradient updates based on the same amount of collected experience by optimizing a surrogate objective, ensuring that the policy distribution does not drift too far from the original one.

We set the discount factor $\gamma$ to 1 to get an unbiased estimate of the expected return. This does not cause the return to diverge, as the rewards are bounded and the trajectory length is constant. We calculate advantages with Generalized Advantage Estimation (Schulman et al., 2015b) with $\lambda = 0.95$.

As a policy network, we use the decoder part of Transformer, without the embedding, so it accepts continuous input. The network has two heads, one predicting the distribution over actions, and the other the value (an estimate of the future return). We add to the PPO objective the L2 loss of the value head with weight 1 and an entropy bonus with weight 0.1.

### 3.2 Model-based approach

As a more sample-efficient, model-based approach, we use Simulated Policy Learning (SimPLe, Kaiser et al. (2017)). The algorithm consists of three phases, repeated in a loop, as shown in Figure 5. First, a set of trajectories is collected, either using the current policy network (which is randomly initialized) or using an external data-set. Then a predictive model of the environment is trained on the collected data. After that, the policy is trained using PPO in the environment simulated by the model. The improved policy is used to collect new data, and the loop continues.

We use a variant of SimPLe where we start the loop from training the model on a set of pre-collected data, and then resume the standard SimPLe algorithm, collecting new data in each iteration.

To be able to use the Transformer decoder as an environment model, we discretize the entire history of observations and actions and concatenate their representations into one long sequence $o_1 a_1 o_2 a_2 ... o_n$. Observations are discretized as described in subsection 2.1. Actions are already discrete, so we just rewrite them using one discrete symbol per each controlled hyperparameter. We train the model to predict just the observation symbols, masking out the loss terms corresponding to action symbols. During infer-

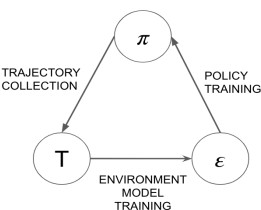

Figure 5: SimPLe: $\pi$ is the policy, $\epsilon$ the environment model, and $T$ the set of collected trajectories.

ence, we calculate the reward based on the difference
of an appropriate metric in two consecutive predicted observations.

For the policy network, we use the same architecture as for the environment model, except for the action and value prediction heads. This lets us pre-initialize the policy parameters with those of the trained environment model, which significantly accelerates the learning process. We conjecture that the attention heads in the policy network needs to attend to similar positions in the sequence as the attention heads in the environment model, so a policy network with parameters initialized in such a way needs to learn mostly the action and value prediction heads, which is much faster.

## 4 RELATED WORK

**Time-Series Modeling with Deep Learning.** Time series forecasting has been traditionally studied using statistical models like ETS (Hyndman et al., 2008) and ARIMA (Box & Jenkins, 1994). There are a number of works that use feed forward neural networks (FFNNs) for time series forecasting, see a survey by Zhang et al. (1998). However, FFNNs break the input series into consecutive fixed size input windows, so the temporal order is ignored within each input window and every new input is considered in isolation. It is also not uncommon to use a hybrid approach of using FFNNs along with ARIMA or ETS as in Khashei & Bijari (2011) and Faruk (2010).

Recurrent Neural Networks (RNNs) are a more natural fit for sequence prediction tasks and have gained popularity for natural language processing tasks. They started to be applied to time-series later and were found to be competitive for point forecasts in a univariate context (Hewamalage et al., 2019). Recently Convolutional Neural Networks (CNNs) have also been used for time series forecasting, especially for capturing long-term dependencies, see van den Oord et al. (2016) and Lai et al. (2018). Other approaches have used Deep Belief Networks and Stacked Denoising Autoencoders to predict temperature and traffic flow (Romeu et al., 2013; Lv et al., 2015). Outside of language and generating audio, Transformers (Vaswani et al., 2017) have not been widely used for time-series prediction.

**Reinforcement Learning.** DQN (Mnih et al., 2013) bootstrapped the field of Deep Reinforcement Learning by training a CNN on raw input pixels to predict the value of future rewards. Further work that built on top of DQN includes Double-DQN by van Hasselt et al. (2016), Dueling-DQN Wang et al. (2016) etc. On the other hand, pure policy optimization techniques like TRPO (Schulman et al., 2015a) and PPO (Schulman et al., 2017) directly represent the policy and optimize the policy parameters. PPO is quite simple to implement and works well on a variety of tasks: Atari, MuJoCo etc. A downside of pure policy gradient algorithms using on-policy learning is the large number of environment interactions needed to achieve satisfactory performance (sample complexity).

**Model-based Reinforcement Learning.** All RL methods mentioned above suffer from the requirement of a large number of interactions with the environment. This can be very costly, as in our case. The idea to use a model of the environment instead of the true one has been explored for a long time. For example, Holland et al. (2018) use a variant of Dyna Sutton (1991) to learn a model of the environment and generate experience for policy training in the context of Atari games. Outside of games, model-based reinforcement learning has been investigated at length for applications such as robotics Deisenroth et al. (2013). Though most of such works do not use image observations, several recent works have incorporated images into real-world robotic control (Finn et al., 2016; Finn & Levine, 2016; Babaeizadeh et al., 2017; Ebert et al., 2017; Piergiovanni et al., 2018; Paxton et al., 2018; Rybkin et al., 2018; Ebert et al., 2018) and simulated Watter et al. (2015); Hafner et al. (2018).

**Hyperparameter Tuning.** Black-box hyperparameter tuning is extremely popular in industry and academia, examples include Google Vizier (Golovin et al., 2017), Hyperopt (Bergstra & Bengio, 2012), Spearmint (Snoek et al., 2012). These approaches assume the existence of a metric on a validation set. Sequential Model-Based Optimization (SMBO), (Hutter et al., 2011) is a family of methods that builds a model of the validation metric with hyperparameters as input, this model is usually trained on the previous trials of the hyperparameters. A large category of SMBO algorithms is Bayesian Optimization that builds a probabilistic model of the above function.A detailed survey about Bayesian Optimization can be found at Shahriari et al. (2016).

As a method of accelerating hyperparameter optimization, Baker et al. (2017) predict the training curve of a network being trained using a deterministic model, and use it for early stopping. We improve upon this work by modeling the stochasticity of training curves and by allowing the hyperparameters

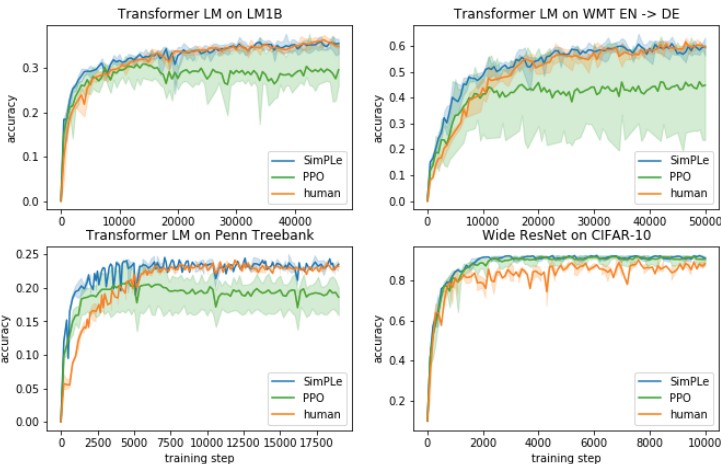

Figure 6: Test accuracy plots for all considered tasks. Each plot shows the minimum, mean and maximum accuracy aggregated over 4 runs of each experiment.

to change over time. Furthermore, we jointly model the training and validation, accuracy and loss curves, while the aforementioned work is restricted to modeling one curve.

Similar to our work, Domhan et al. (2015) and Klein et al. (2017) learn probabilistic models of training curves and use them to accelerate hyperparameter search. Both methods use a handcrafted basis of nonlinear functions of shapes similar to the training curves being modelled. In contrast, our method does not make any assumptions about the shape of the modelled curves, enabling it to capture a wide range of shapes and not requiring the domain knowledge needed to choose such a basis.

Population Based Training of Neural Networks (Jaderberg et al., 2017) is another approach that jointly optimizes the parameters and hyperparameters of a network. It starts off with a population of agents, that after every few training iterations compare their fitness and mutate themselves using traditional evolutionary techniques (copying, mutation etc) and resume training from the next iteration onward, yielding an adaptive hyperparameter schedule that is similar to our work.

Also similar to our work, Xu et al. (2019) learn to tune hyperparameters during model training using reinforcement learning, but they only use a model-free approach and restrict themselves to controlling the learning rate.

Related to our work, Goldblum et al. (2019) draw interesting conclusions about the interaction between learning rate and weight decay rate. They also introduce an alternative to weight decay, which they find to work better in practice.

## 5 EXPERIMENTAL RESULTS

In the experiments, we compare the policy learned using SimPLe with the one learned using model-free PPO and with a manually-tuned baseline. We tune the learning rate, weight decay rate and various other architecture-specific hyperparameters. In all experiments, the manually-tuned baseline changes the learning rate over the course of training and keeps the other hyperparameters constant. This tuning approach, while being simple, is a common practice in the field, as it is infeasible to devise schedules for all hyperparameters manually. The authors of state-of-the-art methods in the considered tasks typically do not report scheduling other parameters than the learning rate (Shoeybi et al. (2019), Radford et al. (2019), Hu et al. (2017)).

For all experiments, we first gather a data-set of 10K training runs of a given architecture on a given task, while varying the hyperparameters using PPO on 128 parallel environments. That means there are 128 concurrent workers training the network in each epoch, after which the policy gets updated and another batch of 128 workers start training. We run 4 parallel runs of this, each for 20 epochs, resulting in $128 * 4 * 20 = 10240$ training curves.

| Architecture and task | SimPLe | PPO | Human |
|---|---|---|---|
| Transformer on LM1B | **35.9%** | 30.2% | 35% |
| Transformer on WMT EN -> DE | **59.9**% | 49.5% | **60%** |
| Transformer on Penn Treebank | **23.4%** | 19.2% | **23.2%** |
| Wide ResNet on CIFAR-10 | **91.6%** | 91.2% | 90% |

Table 1: Final test accuracies for all considered tasks, averaged over 4 runs for each experiment.

On this dataset, we train the forecasting model – a 3-layer Transformer with the loss described in Section 2.1. This forecaster fits a holdout set of curves quite well, see Figure 1. We run 10 epochs of SimPLe to train the policy, in each one gathering 128 additional training curves used to fine-tune the forecaster. In total, we use $10240 + 10 * 128 = 11520$ training curves for each experiment.

We evaluate the learned policies by letting them tune hyperparameters for the corresponding tasks and report the accuracy on the test set (Table 1, Figure 6). We use temperature 0.3 to sample from the policies during evaluation. The resulting policies are significantly better than the ones obtained using actual training curves, as shown in Figure 6, which validates our claim that the forecaster is a useful model of deep learning dynamics.

## 5.1 TRANSFORMER ON LANGUAGE MODELING TASKS

In these experiments we tune the Transformer on 3 language modelling datasets: LM1B, WMT English -> German translation and Penn Treebank. We frame the translation task as a language modeling problem - we concatenate the input and output string with a special symbol in the middle: *input # output* and only use the output string to calculate the loss.

For all tasks we use 6-layer Transformer. In addition to learning rate and weight decay, we also tune 3 dropout parameters: the attention dropout, the dropout in the middle of feed-forward layers and the dropout at the final residual connection in a layer. Not only are there 3 different dropouts, we tune them separately for the first layer, last layer, and all other layers in the middle (2-5). The manually-tuned baseline adjusts the learning rate according to an inverse-square root schedule with a linear warmup, commonly used in the field, and keeps the other hyperparameters constant.

As shown in Figure 6, the SimPLe policy consistently matches or exceeds the performance of the manually-tuned baseline. In all tasks, the SimPLe-tuned model converges significantly faster. The difference gradually decreases during training, with the final performance either better (LM1B) or approximately the same (WMT, PTB) as the baseline. The final accuracies are shown in detail in Table 1. The SimPLe training process is also remarkably stable, yielding similar results across runs of the experiment. We attribute it to the pre-initialization of the policy parameters, which provides a good starting point for policy learning.

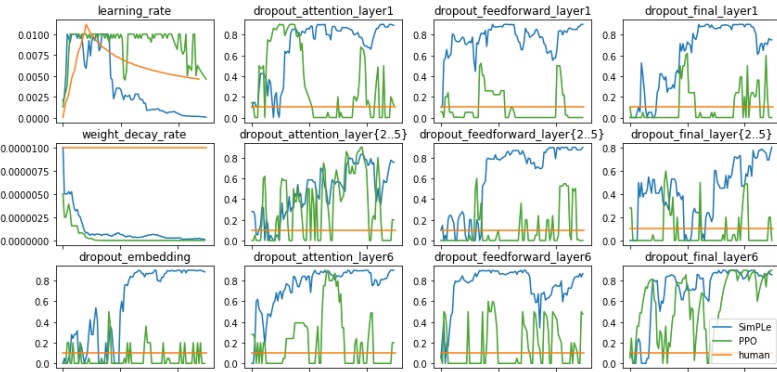

Figure 7: Learned and manually-tuned hyperparameter schedules for Transformer on LM1B.

Figure 8: Learned and manually-tuned hyperparameter schedules for Wide ResNet on CIFAR-10.

In contrast, the PPO training process is quite unstable. The best run of the experiment achieves similar performance as SimPLe on LM1B and WMT, but the mean is significantly worse. Curiously, the PPO-tuned models also converge faster than the manually-tuned ones.

**Discussion.** While the above results attest to the quality of the forecaster, can we draw any insights from the learned policies? We show an example of learned hyperparameter schedules for LM1B in Figure 7. The SimPLe policy clearly employs some interesting strategies. For tuning the learning rate, it follows a well-established pattern of warmup and gradual decay. Dropouts are ramped up to the maximum of 0.9 at varying rates, effectively stopping the learning process for those layers. Interestingly, dropouts of attention layers are increased slower and more gradually than the feedforward layers. The weight decay rate is kept high for a while in the beginning of training and then is quickly decreased. This behavior is also followed by the PPO policy.

## 5.2 WIDE RESNET ON CIFAR-10

In this experiment we tune Wide ResNet (Zagoruyko & Komodakis, 2016) on the CIFAR-10 image classification task. The tuned hyperparameters include the learning rate, weight decay rate and momentum mass. Manually-tuned baseline adjusts the learning rate according to step-wise decrease schedule, commonly used in the field, and keeps the other hyperparameters constant.

In Figure 6 we can see that both the SimPLe and PPO policies outperform the manually-tuned baseline by a large margin. The stability of the PPO training is much higher than in the Transformer experiments, which might be attributed to the smaller action space.

**Discussion.** As shown in Figure 8, the SimPLe policy uses a significantly lower learning rate and higher momentum rate than both the human baseline and PPO policy. Weight decay rate controlled by both SimPLe and PPO follows a similar pattern as in the LM1B task, which suggests that in those tasks weight decay is useful mostly in the initial stage of training.

## 6 CONCLUSIONS

Modeling the learning dynamics of deep models is a complex problem, but a modification of the Transformer model with the discretization strategy we developed in Section 2.1 yields good results in many cases. The predicted curves not only have low L2 distance and look close to the real ones, as seen in Figure 1, but can also be used in to create parameter schedules.

Learning parameter scheduling policies using the forecaster is much more efficient than when running real experiments. A single inference of the forecaster, at batch 128, takes less than a minute even on a single GPU. Training the models on the other hand takes at least an hour and one needs 128 GPUs for that. So it is at least $60 \times 128 = 7680$ times more efficient to operate in simulation. We show experimentally that the forecasting model is indeed good enough to be used in this way.

One question that we leave for future work is how to effectively gather data for the forecaster model. Having a lot of training curves from varied models would allow pre-training the Transformer forecaster before using task-specific data. Pre-training with large data-sets has worked very well, e.g., in the context of BERT and other NLP tasks, so we consider it a promising direction in this case too.

To pre-train the forecaster we will need to scale-up the effort of gathering data. Luckily, this can be a community effort as many people are training models and could benefit from better hyperparameter

schedules. In preparation for this we have already released the code for our experiments as open-source[1] and we are gathering feedback as we extend its applicability.

Finally, aside from the applications to hyperparameter tuning, our forecaster can be applied to any time-series prediction task. We have not experimented with other tasks than the synthetic one, but our results in Section 2.1 are very encouraging and let us believe that Transformer-based models can achieve good results in many time-series prediction tasks.

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
