# OpenReview forum: "Forecasting Deep Learning Dynamics with Applications to Hyperparameter Tuning"
_ICLR.cc/2020/Conference — Reject_

### Official Review · AnonReviewer3 · 2019-10-09
**Official Blind Review #3**

**Rating:** 3

**Review:**

This work focuses on learning a good policy for hyperparameters schedulers, for example learning rate or weight decay, using reinforcement learning. The main contributions include 1) a discretization on the learning curves such that transformer can be applied to predict the them; 2) an empirical evaluations using the predicted learning curves to train the policy.

The main novelties are two folds. On the methodology side, using predicted learning curves instead of real ones can speed up training significantly. On the technical side, the author presented a discretization step to use transformer for learning curve predictions. The results are mixed, we see slightly advantage over human baseline on one task but worse in the other. Human baseline does not need any training! On the writing part, it would be nice to provide more context for both transformer, Proximal Policy Optimization and Simulated Policy Learning to make the paper more self-complete.

I like the directions using surrogate to speed up HPO in general but I feel the learning curve prediction part can be improved. There are already some works, not using deep learning method, for example the following:

* Baker, Bowen, et al. "Accelerating neural architecture search using performance prediction." arXiv preprint arXiv:1705.10823 (2017).
* Domhan, Tobias, Jost Tobias Springenberg, and Frank Hutter. "Speeding up automatic hyperparameter optimization of deep neural networks by extrapolation of learning curves." Twenty-Fourth International Joint Conference on Artificial Intelligence. 2015.

Why these methods are not considered in the beginning? In my opinion, transformer is good for modeling long term dependency and concurrent predictions which is not necessarily the case for learning curves. How does the transformer based method comparing to others?

**Experience Assessment:**

I have read many papers in this area.

**Review Assessment: Checking Correctness Of Derivations And Theory:**

N/A

**Review Assessment: Checking Correctness Of Experiments:**

I assessed the sensibility of the experiments.

**Review Assessment: Thoroughness In Paper Reading:**

I read the paper at least twice and used my best judgement in assessing the paper.

---

> ### Author Response · Authors · 2019-11-15
> **Response to the reviewer's comments**
>
> Thank you for the insightful review.
>
> We updated the paper with better results and more tasks. We show that our method outperforms the human baseline in terms of training speed and either matches or outperforms the human in terms of final accuracy on all tasks. While it is true that the human baseline does not require any additional computational resources for training, it does require domain expertise acquired through years of learning, which is arguably even more costly. Notably, in all 4 problems where we compare to the human baseline, we believe that human researchers used a similar or higher number of runs as our tuner to design the baseline schedules that we compare against.
>
> We also updated the paper with more details regarding Transformer and Proximal Policy Optimization.
>
> Thank you for mentioning the existing learning curve modeling methods. We added an explanation of differences of our method with those works. [1] learn a probabilistic model of one training curve using a handcrafted basis of nonlinear functions of shapes similar to the training curves being modelled. Our method does not make any assumptions about the shape of the modelled curves and is able to jointly model many training curves - in our experiments, training and validation loss and accuracy. [2] learn a deterministic model of a learning curve, while our method also models stochasticity, hence providing diverse experience for training a reinforcement learning agent. Also in contrast to [1] and [2], our method allows the hyperparameters to change over the course of training and models the influence of those changes on the training metrics.
>
> [1] Baker, Bowen, et al. "Accelerating neural architecture search using performance prediction." arXiv preprint arXiv:1705.10823 (2017).
>
> [2] Domhan, Tobias, Jost Tobias Springenberg, and Frank Hutter. "Speeding up automatic hyperparameter optimization of deep neural networks by extrapolation of learning curves." Twenty-Fourth International Joint Conference on Artificial Intelligence. 2015.

---

### Official Review · AnonReviewer1 · 2019-10-23
**Official Blind Review #1**

**Rating:** 6

**Review:**

The paper investigates the possibility of learning a model to predict the training behaviour of deep learning architectures from hyperparameter information and a history of training observations. The model can then be used by researchers or a reinforcement learning agent to make better hyperparameter choices. The paper first adapts the Transformer model to be suitable to this prediction task by introducing a discretization scheme that prevents the transformer decoder's predictions from collapsing to a single curve. Next, the problem is formalized as a partially-observable MDP with a discrete action set, and PPO and SimPLe are introduced. The proposed model-based method is compared against a human and a model-free baseline training a Wide ResNet on CIFAR-10. The model-based method achieves better validation error than the other baselines that use actual data. Next, the method is compared against a human and a model-free baseline training Transformer models on the Penn Treebank dataset. While the human achieves the best performance at the end of the run, the proposed method appears to learn more quickly than the others and finishes with performance comparable to the model-free baseline.

Currently I lean towards accepting this paper for publication, despite a few issues. It asks an interesting question: can we learn a model of the training dynamics to avoid actually having to do the training? This could potentially prevent a lot of unnecessary computation and also lead to better-performing models. It then shows some experimental evidence suggesting that this is possible.

Most importantly, I would like to see a measure of variance/uncertainty like confidence intervals included in the results; otherwise it's impossible to assess whether the results are likely to be significant or not. Other questions:
1. In the PTB experiment, it looks like the human only adapts the learning rate and leaves the rest of the hyperparameters alone. Why was this policy used as the baseline? It seems extremely basic and unlikely to truly lead to optimal performance.
2. Why were more baselines from the related work not included? I understand the experiments are a proof of concept, but it would be nice to get a feeling for what some of the other methods do.
3. How do PPO and SimPLe handle partial observability? Is it principled to apply them to partially-observable environments?
4. Why not use continuous actions with a parameterized policy (e.g. Gaussian)?
5. Is it reasonable to assume that the learning dynamics of all deep learning architectures are similar enough that a model trained on one set of deep learning architectures and problems will generalize to new architectures and problems?

**Experience Assessment:**

I do not know much about this area.

**Review Assessment: Checking Correctness Of Derivations And Theory:**

N/A

**Review Assessment: Checking Correctness Of Experiments:**

I assessed the sensibility of the experiments.

**Review Assessment: Thoroughness In Paper Reading:**

I read the paper at least twice and used my best judgement in assessing the paper.

---

> ### Author Response · Authors · 2019-11-15
> **Response to the reviewer's comments**
>
> We thank the reviewer for their comprehensive review.
>
> We updated the paper with better results over more tasks, either matching or outperforming the human baseline in terms of final accuracy, and outperforming the model-free baseline in all cases. We also included results over multiple runs of all experiments, showing the minimum, maximum and mean accuracy.
>
> 1. While it is true that the manually-tuned baseline we provided is simple, it is a standard practice in the field to adjust the learning rate during training and keep the rest of the hyperparameters constant. Adjusting all of them requires significantly more effort and is infeasible in many cases.
>
> 2. Due to time constraints, we have not benchmarked our method against more hyperparameter-tuning baselines yet. We agree that it would be a very valuable comparison and leave that for future work. Nevertheless, please note that the human baselines we use for Transformer have been tuned by researchers using auto-tuners among other tools.
>
> 3. [1] successfully use PPO with an LSTM policy on a challenging, partially-observable environment. It is equally principled to use a Transformer policy, since both would operate on the same sequence of observations. The SimPLe algorithm runs PPO on an MDP approximated by a powerful model that handles stochasticity well, which is also a valid approach.
>
> 4. We updated the paper with a justification of our action discretization scheme. Such a discretization has a number of benefits, including multi-modality, which cannot be achieved using a parameterized Gaussian policy. [2] show that discretization of the action space improves the average performance, stability and robustness to hyperparameters of reinforcement learning agents on a range of continuous control tasks.
>
> 5. While we have not included such transfer experiments in our current work, we do believe that a model trained on enough architectures and tasks will generalize to new ones. For instance, in the updated version of the paper, we show that the learned policy employs similar learning rate and weight decay rate adjustment schemes across very different tasks. Substantiating this claim in the general case will likely require a large-scale study, which we plan to perform in the future.
>
> [1] OpenAI et al. “Learning Dexterous In-Hand Manipulation”, arXiv preprint arXiv:1808.00177 (2018)
>
> [2] Tang et al. “Discretizing Continuous Action Space for On-Policy Optimization”, arXiv preprint 1901.10500 (2019)

---

> > ### Comment · AnonReviewer1 · 2019-11-15
> > **Replying to the authors' response**
> >
> > Thank you for reading my review and responding to my comments.

---

### Official Review · AnonReviewer2 · 2019-10-23
**Official Blind Review #2**

**Rating:** 1

**Review:**

This paper proposed to train a network with training curves and corresponding parameters, and use policy search to find optimal parameter to replace hundreds or thousands of training in real case scenario, and it is clearly much faster using the trained network to infer parameters, instead of tuning the network manually.

The first point would be: what's the meaning of synthetically generating training curves other than proving that transformer achieves good performance in modeling discrete distribution? Most practical problems would not have the same distribution as the previously gathered public dataset, thus the data is not representative, and synthetic training curves just does not make sence.

The cited paper 'Learning an adaptive learning rate schedule' does not appear online.



**Experience Assessment:**

I do not know much about this area.

**Review Assessment: Checking Correctness Of Derivations And Theory:**

I assessed the sensibility of the derivations and theory.

**Review Assessment: Checking Correctness Of Experiments:**

I assessed the sensibility of the experiments.

**Review Assessment: Thoroughness In Paper Reading:**

I made a quick assessment of this paper.

---

> ### Author Response · Authors · 2019-11-15
> **Response to the reviewer's comments**
>
> We thank the reviewer for the effort, however we believe there is a mis-understanding.
>
> As for the synthetic curves experiment, we updated the paper with a justification. This task, while simple, showcases the ability of Transformer to model a distribution over curves of similar shape to real training curves with varying speeds of convergence. It has been designed so it is easy to quantify the diversity of generated curves and the fit between the distribution generated by the model and the real one. Furthermore, we included two additional tasks, attesting to the ability of Transformer to model a wide range of distributions over training curves. We also updated the citation of the paper you mentioned with an arxiv URL.
>
> We still believe that while focusing on the synthetic task the reviewer might have missed the main point of the paper, namely that time-series forecasting with Transformer works really well, at least in the context of modeling deep learning dynamics. The general problem has been studied in the community for many decades and we believe that we made significant progress, so we kindly encourage the reviewer to reconsider their assessment of our contributions.

---

### Public Comment · ~Micah_Goldblum1 · 2019-11-08
**An Interesting Connection**

Hi Authors,
Thank you for your interesting paper.  I wanted to bring to your attention that your insights into learning rate and weight decay is related to our paper, which shows that an alternative to weight decay may stabilize effective learning rate and can improve performance.[1]  Please consider mentioning the relationship with our work in your next version.

[1] https://arxiv.org/abs/1910.00359

---

> ### Author Response · Authors · 2019-11-15
> **Response to the comment**
>
> Thank you for mentioning this connection. We have added it to the updated version of our work.

---

### Decision · Program_Chairs · 2019-12-19

**Decision:**

Reject

**Comment:**

This paper trains a transformer to extrapolate learning curves, and uses this in a model-based RL framework to automatically tune hyperparameters. This might be a good approach, but it's hard to know because the experiments don't include direct comparisons against existing hyperparameter optimization/adaptation techniques (either the ones based on extrapolating training curves, or standard ones like BayesOpt or PBT). The presentation is also fairly informal, and it's not clear if a reader would be able to reproduce the results. Overall, I think there's significant cleanup and additional experiments needed before publication in ICLR.